# Ammonia in the UTLS: GLORIA airborne measurements for CAMS model evaluation in the Asian Monsoon and in biomass burning plumes above the South Atlantic

Sören Johansson[1], Michael Höpfner[1], Felix Friedl-Vallon[1], Norbert Glatthor[1], Thomas Gulde[1], Vincent Huijnen[2], Anne Kleinert[1], Erik Kretschmer[1], Guido Maucher[1], Tom Neubert[3], Hans Nordmeyer[1], Christof Piesch[1], Peter Preusse[4], Martin Riese[4], Björn-Martin Sinnhuber[1], Jörn Ungermann[4], Gerald Wetzel[1], and Wolfgang Woiwode[1]

[1]Institute of Meteorology and Climate Research - Atmospheric Trace Gases and Remote Sensing (IMK-ASF), Karlsruhe Institute of Technology, Karlsruhe, Germany
[2]R&D Weather and Climate models, Royal Netherlands Meteorological Institute (KNMI), De Bilt, Netherlands
[3]Central Institute of Engineering, Electronics and Analytics - Electronic Systems (ZEA-2), Forschungszentrum Jülich, Jülich, Germany
[4]Institute of Energy and Climate Research - Stratosphere (IEK-7), Forschungszentrum Jülich, Jülich, Germany

**Correspondence:** S. Johansson (soeren.johansson@kit.edu)

**Abstract.** Ammonia ($NH_3$) is the major alkaline species in the atmosphere and plays an important role in aerosol formation, which affects local air quality and the radiation budget. $NH_3$ in the Upper Troposphere and Lower Stratosphere (UTLS) is difficult to detect and only limited observations are available. We present two dimensional trace gas measurements of $NH_3$ obtained by the airborne infrared imaging limb sounder GLORIA (Gimballed Limb Observer for Radiance Imaging of the Atmosphere) that has been operated onboard the research aircraft Geophysica within the Asian Monsoon Anticyclone during the StratoClim campaign (July 2017) and onboard HALO (High Altitude and Long Range Research Aircraft) above the South Atlantic during the SouthTRAC campaign (September-November 2019). We compare these GLORIA measurements in the UTLS with results of the CAMS (Copernicus Atmosphere Monitoring Service) reanalysis and forecast model to evaluate its performance. The GLORIA observations reveal large enhancements of $NH_3$ of more than 1 ppbv in the Asian Monsoon upper troposphere, but no clear indication of $NH_3$ in biomass burning plumes in the upper troposphere above the South Atlantic above the instrument's detection limit of around 20 pptv. In contrast, CAMS reanalysis and forecast simulation results indicate strong enhancements of $NH_3$ in both measured scenarios. Comparisons of other retrieved pollution gases, such as peroxyacetyl nitrate (PAN), show the ability of CAMS models to generally reproduce the biomass burning plumes above the South Atlantic. However, $NH_3$ concentrations are largely overestimated by the CAMS models within these plumes. We suggest that emission strengths used by CAMS models are of lower accuracy for biomass burning in comparison to agricultural sources in the Asian Monsoon. Further, we suggest that loss processes of $NH_3$ during transport to the upper troposphere may be underestimated for the biomass burning cases above the South Atlantic. Since $NH_3$ is strongly undersampled, in particular at higher altitudes, we hope for regular vertically resolved measurements of $NH_3$ from the proposed CAIRT mission to strengthen our understanding of this important trace gas in the atmosphere.

## 1 Introduction

Ammonia ($NH_3$) is the major alkaline trace gas of the atmosphere and part of total reactive nitrogen. Sources of atmospheric $NH_3$ are livestock and fertilizer in agriculture, but also industry and combustion processes (e.g., Bouwman et al., 1997). Further, $NH_3$ has been measured in biomass burning plumes in vicinity of fires (e.g., Hegg et al., 1988; Coheur et al., 2007; Tomsche et al., 2023). It is expected that total emissions of $NH_3$ rise strongly, due to increased livestock, usage of fertilizers, and combustion (Szopa et al., 2021, and references therein). Atmospheric sinks of $NH_3$ are wash-out due to the high water solubility of $NH_3$, and formation of aerosols, like ammonium sulfate (in presence of sulfuric acid) or ammonium nitrate (in presence of nitric acid). The importance of $NH_3$ in the atmosphere for the initial formation of new particles has been corroborated by various studies (e.g., Kirkby et al., 2023, and references therein). As aerosols are of importance for local air quality, and for climate through their direct and indirect radiative impact (e.g., Szopa et al., 2021), it is necessary to observe and understand the distribution of $NH_3$ in the atmosphere from the boundary layer up to the lower stratosphere.

In situ measurements of $NH_3$ are, however, challenging due to the wide range of ambient levels (5 pptv to 500 ppbv) and the interaction of $NH_3$ with surfaces of the instrument ("sticky"; e.g., von Bobrutzki et al., 2010). Still, there are various in-situ measurements at boundary layer altitudes, but only very few in the free troposphere and very sparse measurements with high uncertainties in the upper troposphere (e.g., Ziereis and Arnold, 1986; Nowak et al., 2010; Tomsche et al., 2023) .

Due to its specific spectral lines in the thermal infrared region, remote sounding spectrometers are suited to quantify $NH_3$ in a "contact-free" manner. In that way, $NH_3$ has been observed by infrared sounders from satellite and aircraft in nadir and limb geometry. From satellite instruments measuring in nadir, such as TES (Tropospheric Emission Spectrometer), IASI (Infrared Atmospheric Sounding Interferometer), CrIS (Cross-track Infrared Sounder), or AIRS (Atmospheric Infrared Sounder), global observations of $NH_3$ in the lower troposphere are available (e.g., Beer et al., 2008; Clarisse et al., 2010; van Damme et al., 2015; Shephard and Cady-Pereira, 2015; Warner et al., 2016). First observations of $NH_3$ in the upper troposphere (in particular of the Asian summer monsoon) have been achieved by the MIPAS (Michelson Interferometer for Passive Atmospheric Sounding) instrument from the Envisat satellite (Höpfner et al., 2016), and by the GLORIA (Gimballed Limb Observer for Radiance Imaging of the Atmosphere) instrument onboard the Geophysica research aircraft (Höpfner et al., 2019). The connection of large amounts of $NH_3$ reaching the upper troposphere through convection with the formation of solid ammonium nitrate (AN) particles as secondary aerosols was also shown during these flights (Höpfner et al., 2019; Wagner et al., 2020; Appel et al., 2022).

In the present work, we utilize observations of mid- to upper tropospheric $NH_3$ obtained within two aircraft campaigns of the GLORIA instrument, to evaluate the ability of the CAMS atmospheric model to describe the distribution of $NH_3$ under two different scenarios: within the Asian summer monsoon and within biomass burning plumes. Beside the direct comparison of $NH_3$, we further use peroxyacetyl nitrate (PAN) as a long-lived ($\tau \approx 3$ months) pollution tracer, which has been shown to be useful to identify biomass burning plumes in GLORIA and CAMS data (e.g. Johansson et al., 2022).

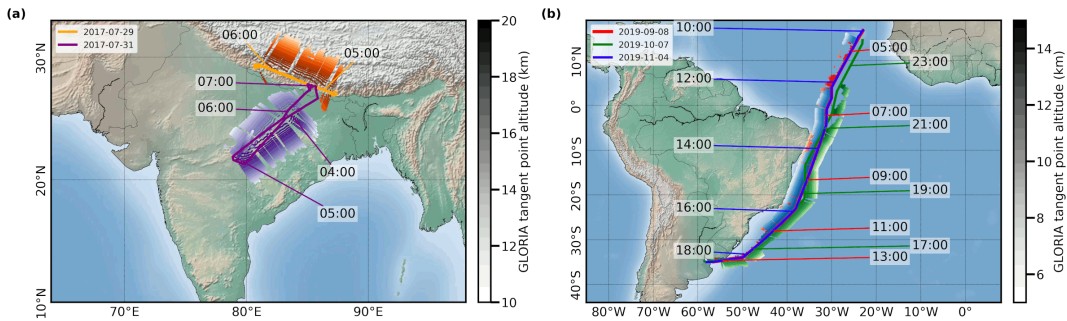

**Figure 1.** Flight paths and locations of GLORIA tangent points for discussed flights of (a) the StratoClim campaign and (b) the SouthTRAC campaign. Tangent points affected by clouds are not used in the retrieval process and result in gaps in the plotted geolocations. Note different color bar ranges and map scales for each panel. The flight path of the flight on 8 September 2019 is identical to the flight path of the flight on 4 November 2019, which hides large parts of the earlier flight.

## 2 Observations and atmospheric model simulations

### 2.1 GLORIA measurements

The GLORIA instrument has been deployed on various airborne campaigns with the M55 Geophysica and the HALO research
aircraft. In this study, GLORIA measurements from the M55 Geophysica StratoClim campaign (July/August 2017, base in
Kathmandu, Nepal), and from the HALO SouthTRAC campaign (September-November 2019 with bases in Oberpfaffenhofen,
Germany and Rio Grande, Argentina) are discussed. Flight paths and the location of GLORIA tangent points are summarized
in Fig. 1. For this study, StratoClim flights on 29 July 2017 and 31 July 2017 have been selected because of measured high
$NH_3$ (Höpfner et al., 2019), while SouthTRAC flights on 8 September 2019, 7 October 2019, and 4 November 2019 are chosen
because of measured large pollution plumes above the South Atlantic (Johansson et al., 2022). Both StratoClim flights were
conducted from and to Kathmandu, Nepal, and the three selected SouthTRAC flights were directed from Sal, Cape Verde, to
Buenos Aires, Argentina, and vice versa, as part of the transfer flights between Germany and Argentina.

The GLORIA instrument (Friedl-Vallon et al., 2014; Riese et al., 2014) combines an imaging detector working in the
thermal infrared with a Fourier-Transform-Spectrometer in an actively controlled gimbal frame. This combination allows for
simultaneous observations of $128 \times 48$ (vertical $\times$ horizontal) atmospheric spectra, high spectral sampling up to $0.0625 \, cm^{-1}$,
and compensation of aircraft movements and active targeted line-of-sight control. Further, two external black bodies are used
for in-flight radiometric calibration measurements, together with upward looking atmospheric "deep space" observations. For
the measurements discussed in this work, interferograms which were obtained with a maximum optical path difference (MOPD)
of 8.0 cm (spectral sampling of $0.0625 \, cm^{-1}$) are used. Only during the StratoClim flight on 29 July 2017, due to operational
restrictions, interferograms with 2.5 cm MOPD were recorded, resulting in a spectral sampling of $0.2 \, cm^{-1}$. Within GLORIA
level 1 processing, the interferograms are Fourier-Transformed and radiometrically and spectrally calibrated. The resulting
calibrated spectra at each detector pixel are screened, filtered, and finally horizontally binned into 127 spectra per recorded

measurement sampled in the vertical direction (with one horizontal line of spectra beeing disregarded completely due to detector artifacts; Kleinert et al., 2014; Ungermann et al., 2022). All GLORIA retrievals in this study are based on version

v03.02 of GLORIA level 1 data (calibrated spectral radiances).

Based on these binned calibrated spectra, profiles of atmospheric temperature and trace gases are retrieved perpendicular to the flight track (Johansson et al., 2018). The retrieval uses a nonlinear least-squares fit with Tikhonov regularization of the retrieval software KOPRAFIT together with the radiative transfer model KOPRA (Stiller, 2000). For $NH_3$, the overall retrieval strategy from Höpfner et al. (2019) is applied, for PAN, retrievals as described by Johansson et al. (2020, 2022) are utilized,

and for AN, retrievals as described by Höpfner et al. (2019). Due to the reduced spectral sampling on the flight of 29 July 2017, spectral windows and regularization have been adjusted slightly (see supplementary Tab. S1). For $NH_3$ retrievals of StratoClim flights, vertical resolutions between 0.8 and 1.3 km, and estimated errors between 10 and 50 pptv have been achieved, for SouthTRAC flights discussed in this work, vertical resolutions between 0.5 and 0.7 km, and estimated errors between 6 and 12 pptv (10,90 percentile ranges, each).

In addition, for cloud information, the so-called cloud index is calculated as a ratio of spectral windows (see Spang et al., 2004). Please note that lower cloud index values mean a higher contamination of clouds along the line of sight.

## 2.2 CAMS global atmospheric composition model configurations

As part of the Copernicus Atmosphere Monitoring Service (CAMS), two data products are publicly available: First, the ECMWF (European Centre for Medium-Range Weather Forecast) Atmospheric Composition Reanalysis version 4 (EAC4;

Flemming et al., 2015; Inness et al., 2019), and second, the CAMS global atmospheric composition operational near-real time forecasts. Both atmospheric model configurations use the ECMWF Integrated Forecast System (IFS) model, and assimilate multiple observations of atmospheric state and composition. In this study, CAMS model output is linearly interpolated in space and time onto GLORIA measurement geolocations to ensure optimal comparisons between observation and model output.

For this work, we compare GLORIA data to both CAMS configurations to achieve two goals: First, we want to compare

GLORIA $NH_3$ obtained during two different campaigns 2017 and 2019 with a consistent data set that did not change for the time ranges of both campaigns. For this purpose, the reanalysis configuration is suited best. Second, we want to compare GLORIA $NH_3$ to the most recent CAMS configuration that is publicly available, which is the CAMS forecast configuration, in order to check for improvements for more recent model versions.

### 2.2.1 CAMS reanalysis

The CAMS reanalysis is currently available between 2003 and 2021, uses 60 vertical levels between 0.1 and 1000 hPa, and a horizontal resolution of $0.75° \times 0.75°$ latitude $\times$ longitude. Data is provided every 3 h and includes meteorological parameters, concentrations of chemical trace gases, as well as aerosols. It is based on IFS cycle 42r1, and employs data assimilation of trace gases $O_3$, CO, and $NO_2$, along with aerosol optical depth (Inness et al., 2019). Chemistry is handled by a module named IFS(CB05) (Flemming et al., 2015), with its tropospheric chemistry as inherited from the TM5 model (Huijnen et al., 2010),

and aerosols are treated as described by Morcrette et al. (2009). Anthropogenic emissions are prescribed from MACCity

(MACC/CityZEN; Granier et al., 2011), biogenic emissions from MEGAN2.1 (Model of Emissions of Gases and Aerosols from Nature; Guenther et al., 2012), and biomass burning emissions from GFAS v1.2 (Global Fire Assimilation System; Kaiser et al., 2012). CAMS reanalysis has been evaluated by various aircraft measurements: Wang et al. (2020) compared tropospheric trace gas profiles over the Arctic, North America, and Hawaii to CAMS reanalysis and show that simulated PAN is in agreement with observations. Further, Johansson et al. (2020, 2022), and Wetzel et al. (2021) compared GLORIA PAN and other pollution trace gases to CAMS reanalysis data in the upper troposphere above the Asian Monsoon, the North Atlantic, and the South Atlantic, respectively. They found an overall agreement for the measured plume structures and for biomass burning plumes above the South Atlantic (including SouthTRAC flights that are discussed in this study). In these plumes especially the trace gas PAN was even in quantitative terms described well by the model. To our knowledge, there are no studies available evaluating $NH_3$ for CAMS reanalysis. However, since $NH_3$ and PAN can be emitted both from biomass burning events, it is important to know that CAMS is able to reproduce biomass burning PAN plumes.

### 2.2.2 CAMS forecast

The CAMS global atmospheric composition near-real time forecast is operationally available since 2015, and in contrast to the reanalysis, this data product receives a version upgrade approximately once a year. For this reason, the CAMS forecast data for StratoClim measurements in 2017 (cycle 43r1) are different to those for SouthTRAC measurements in 2019 (cycle 46r1). Both forecast model results are based on a newer model version compared to CAMS reanalysis (cycle 42r1). The model utilizes 60 vertical levels for simulations for 2017, and 137 model levels for 2019, and in both cases a horizontal resolution of $0.4° \times 0.4°$ latitude $\times$ longitude. New forecasts are started every day at 0:00 UTC and 12:00 UTC, and output is provided every 3 h for vertically resolved parameters. In this work, model output of shortest possible lead times of 0, 3, 6, and 9 h are used for interpolation as mentioned above.

Besides changes in resolution, CAMS forecast for the measurements in 2017 differs compared to CAMS reanalysis, among other things in the following aspects: Additional observational data sets are considered for assimilation (e.g., vertical ozone profiles of the Ozone Mapping and Profiler Suite). Further, a new source scheme for Secondary Organic Aerosols, which is based on scaled CO emissions, is introduced (Eskes et al., 2016). Additionally, CAMS forecasts for the measurements in 2019 differs among other things in these aspects: Now, CAMS makes use of `CAMS_GLOB_ANT` v2.1 anthropogenic and `CAMS_GLOB_BIO` v1.1 biogenic emissions. Biomass burning plume injection height from GFAS is introduced, along with a diurnal cycle in its emissions. Further, separate ammonium and nitrate aerosol species are introduced, interaction between chemistry and aerosol schemes are established, which implied a change in the modelled $NH_3$ life cycle. Also a selection of chemical reaction rates are updated (Eskes et al., 2017, 2018; Basart et al., 2019).

In order to compare CAMS forecast ammonium aerosol to GLORIA AN, the CAMS ammonium aerosol mass mixing ratio is scaled according to the molar mass ratio of ammonium nitrate to ammonium, and then converted to mass density. This approach may overestimate AN if not all of the ammonium aerosol in the model is present in the form of AN.

## 3 Ammonia measurements and comparisons

In this section, GLORIA measurements of $NH_3$ are presented and compared to CAMS reanalysis and forecast simulation results. In addition, as indicator for the pollution plumes, GLORIA measurements of PAN are shown. First, the two flights from the StratoClim campaign within the Asian monsoon air are presented, followed by three SouthTRAC flights concentrating on biomass-burning situations.

### 3.1 Asian Monsoon

From space-borne infrared nadir and limb-observations it is known that elevated concentrations of $NH_3$ are present in the atmosphere above the Indian subcontinent at lower levels (van Damme et al., 2015) and even in the upper troposphere during the time of the Asian Monsoon (Höpfner et al., 2016). For the GLORIA airborne measurements on 31 July 2017, strongly enhanced abundances of $NH_3$ of up to about 1000 pptv were reported (Höpfner et al., 2019). The current analysis of the flight on 29 July 2017 above Nepal shows even higher concentrations of up to 1500 pptv, as presented in Fig. 2a. This maximum is part of a larger horizontal structure between 5:30 UTC and 6:15 UTC and 12 km to 14 km altitude. Peak values of $NH_3$ VMRs coincide with maximum PAN VMRs (Fig. 2d). CAMS reanalysis (Fig. 2b) indicates a similar $NH_3$ enhancement at the same altitude range, but shifted to earlier measurement times (eastwards). In addition, a thin maximum up to 700 pptv $NH_3$ is present at 15 km altitude, which is at a similar location as a second, smaller observed PAN maximum (see Fig. 2d), but does not correspond to a measured $NH_3$ maximum. CAMS forecast also shows two plumes, but spatially shifted and of different intensity. This model indicates a major $NH_3$ maximum around 6:00 UTC peaking at approximately 14 km altitude. Maximum VMRs are up to 900 pptv and thus smaller than observed, which may be caused by too fast $NH_4^+$ production in the model. Another maximum is forecasted at approximately 05:30 UTC and 12 to 16 km altitude with maximum VMRs up to 1000 pptv at 12 km altitude.

The StratoClim flight on 31 July 2017 reveals one large enhancement of $NH_3$, peaking at 4:15 UTC and 14 km altitude with VMRs up to 1000 pptv (Fig. 3a). A second, faint maximum (4:30-4:50 UTC, 12 km altitude) indicates lower VMRs up to 100 pptv. For the second part of the flight (later than 5:00 UTC), GLORIA measurements show no enhanced $NH_3$. This contrast in $NH_3$ distributions between both parts of the flight is remarkable, since the flight region of the second part of the flight is relatively close to the flight region of the first part of the flight, and mainly the viewing direction of GLORIA has changed from south-eastward to north-westward (see Fig. 1a). Hence, our GLORIA measurements indicate strong horizontal variability of $NH_3$ in the Asian Monsoon upper troposphere. However, PAN observations indicate polluted air masses also for this second part of the flight. These PAN pollution plumes in the second part of the flight are reproduced by CAMS reanalysis and forecast (see Johansson et al., 2020 and supplementary Fig. S1 and Fig. S2). CAMS reanalysis and forecast successfully reproduce the approximate position of the observed $NH_3$ plumes, as well as the absence of $NH_3$ plumes in the second part of the flight (see Fig. 3b,c). However, CAMS reanalysis does not simulate the major plume later than 4:10 UTC, and also maximum VMRs are simulated below 400 pptv, and thus considerably lower than the measured ones. CAMS forecast also simulates the largest VMRs of the major plume earlier than measured, but maximum VMRs are comparable to the GLORIA measurements.

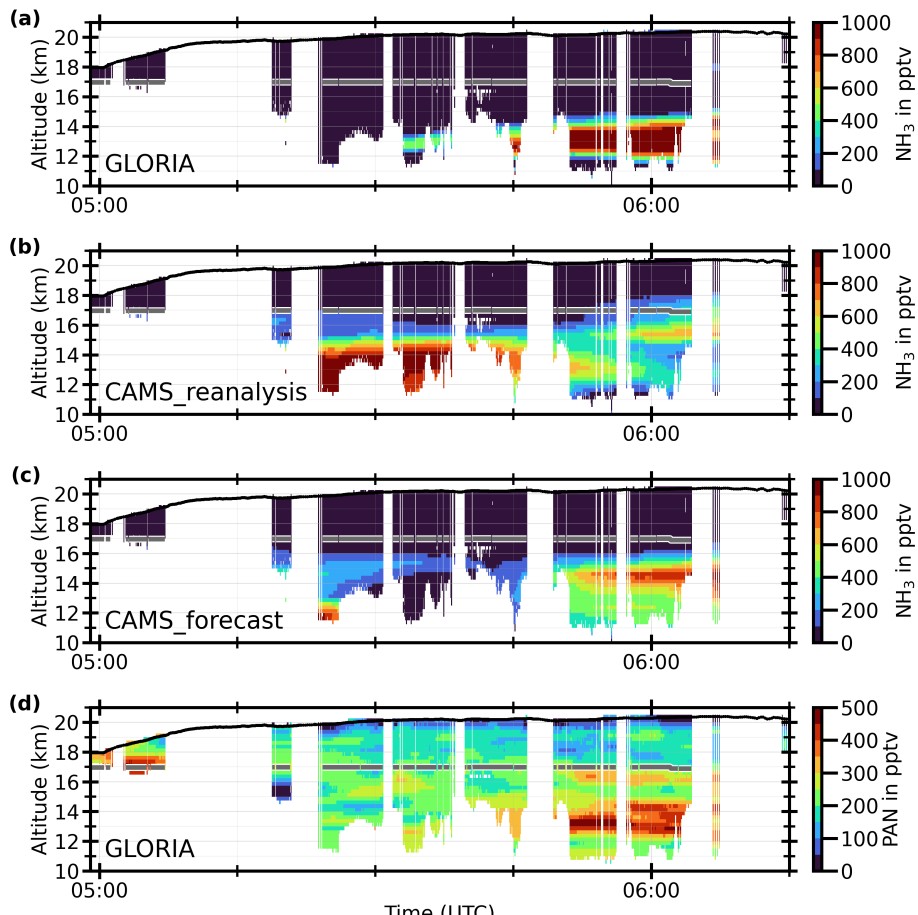

**Figure 2.** StratoClim flight on 29 July 2017 from and to Kathmandu, Nepal: GLORIA time/altitude cross sections of (a) NH$_3$ and (d) PAN together with CAMS reanalysis (b) and CAMS forecast (c) simulation results, interpolated onto GLORIA geolocations. GLORIA data is horizontally averaged to match lower horizontal resolutions of CAMS forecast of ≈44 km. The black line indicates flight altitudes, the gray line shows the ECMWF analysis 380 K potential temperature as indication of the tropopause location in the Asian Monsoon. Blank spaces indicate regions of high cloud tops, calibration measurements, or aircraft movements (like curves). Maximum NH$_3$ VMRs are measured up to 1500 pptv and thus exceed color bar maximum.

## 3.2 Biomass burning plumes above the South Atlantic

All three of the transfer flights between Sal, Cape Verde, and Buenos Aires, Argentina, and vice versa, have been influenced by pollution plumes, originating from biomass burning events (Johansson et al., 2022). In the following we will discuss the observations of NH$_3$ by GLORIA within these plumes in comparison to both CAMS reanalysis and forecast model configurations.

For the SouthTRAC flight on 8 September 2019 (Fig. 4a), GLORIA measured less than 20 pptv NH$_3$ over large parts of the flight, despite clear biomass burning plume structures observed in PAN (Fig. 4d). Very small enhancements below 60 pptv

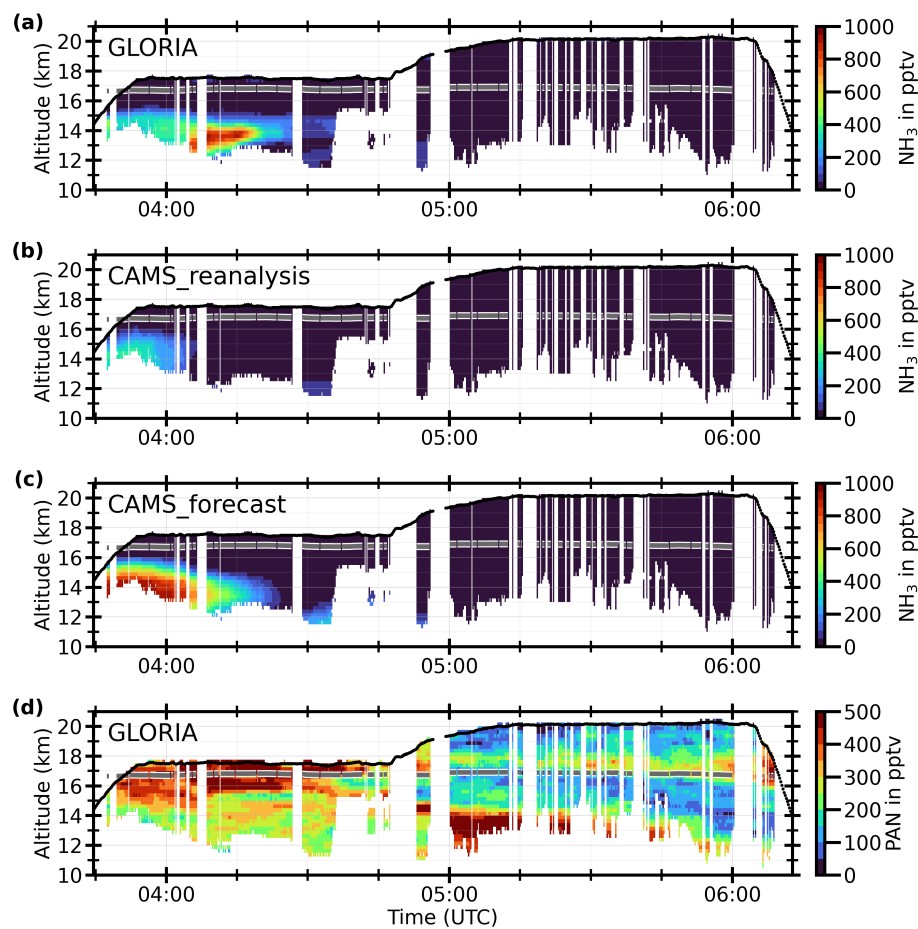

**Figure 3.** Same as Fig. 2, but for flight on 31 July 2017. Maximum $NH_3$ VMRs are measured up to 1100 pptv and thus exceed color bar maximum.

are indicated at approximately 10:15 UTC and 13 km altitude, surrounding the filtered air masses, where no retrieval has been possible due to cloud or aerosol contamination. In such close proximity, the cloud or aerosol feature, however, could have affected the $NH_3$ retrieval quality. CAMS reanalysis (Fig. 4b) and forecast (Fig. 4c) simulate enhanced concentrations of $NH_3$

180 within extended areas, which are correlated with the PAN biomass burning plumes. CAMS reanalysis shows two horizontal plumes with VMRs up to 250 pptv, between 9:30 and 11:00 UTC and at 8 and 11 km altitude. This structure corresponds to the measured biomass burning plumes. Using backward trajectories, it has been shown that these air masses likely entered the upper troposphere above central South America (see Fig. 3 of Johansson et al., 2022), while other air masses with enhanced PAN originated from Africa. Further, GLORIA measured $C_2H_4$ (upper tropospheric life time of $\approx$1.2 d) exclusively at these

185 spots, which indicates a young age of these air masses (see Fig. 1 of Johansson et al., 2022). Similarly, CAMS forecast indicates maximum VMRs up to 400 pptv for these plumes. In addition, CAMS forecast also simulates enhancements of lower VMRs up

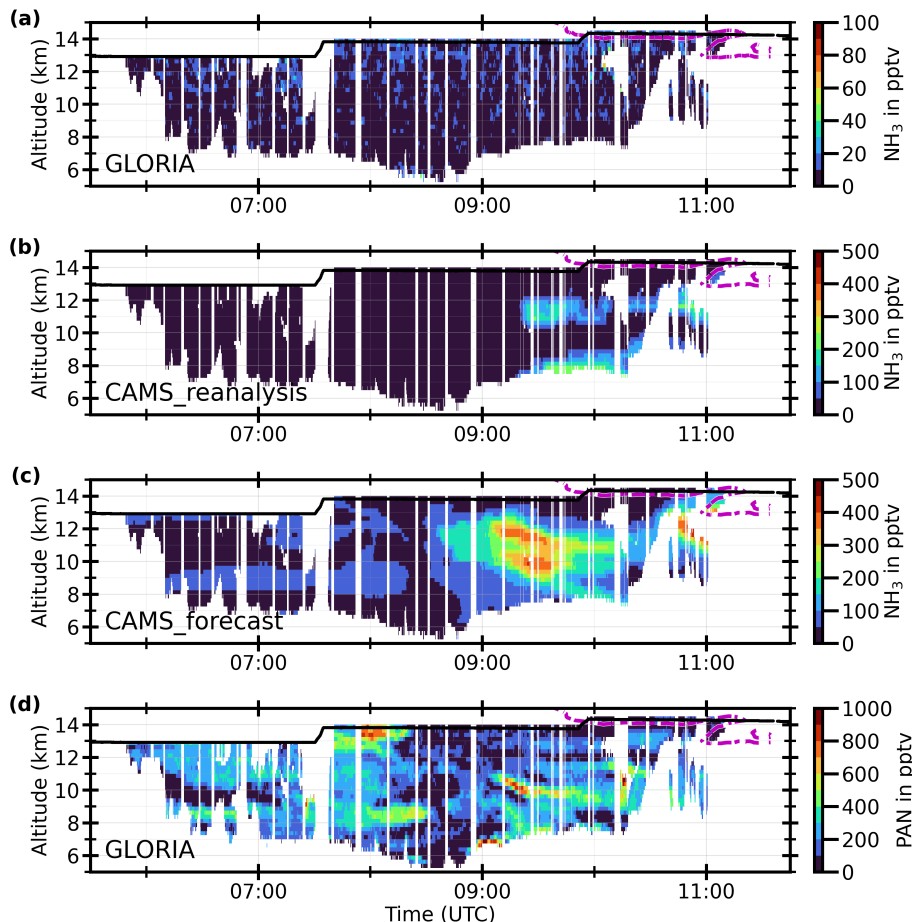

**Figure 4.** SouthTRAC flight on 8 September 2019 from Sal, Cape Verde, to Buenos Aires, Argentina: GLORIA time/altitude cross sections of (a) $NH_3$ and (d) PAN together with CAMS reanalysis (b) and CAMS forecast (c) simulation results, interpolated onto GLORIA geolocations. GLORIA data is horizontally averaged to match lower horizontal resolutions of CAMS forecast of $\approx 44$ km. The black line indicates flight altitudes, the dashed magenta line shows the ECMWF $\pm 2, 4$ PVU isolines as indication of the tropopause. Due to tropopause heights, which are partially located above flight altitude, these magenta lines are not always present, and sometimes only the $\pm 2$ PVU line is present. Blank spaces indicate regions of high cloud tops, calibration measurements, or aircraft movements. Note that color bars for $NH_3$ change between GLORIA measurements (a) and both CAMS simulation results (b-c). Further, color bars have been adjusted to generally lower $NH_3$ VMRs compared to Figs. 2-3.

to 100 pptv for air masses associated with PAN plumes in the first part of the flight (e.g., 6:15 to 8:30 UTC and 9 km altitude, and 8:00 UTC and 12 km altitude). No similar enhancements of $NH_3$ are visible in the GLORIA observations.

The SouthTRAC flight on 7 October 2019 was characterized by a large, nose-like plume of PAN (and other pollution species) with high VMRs up to 1000 pptv (17:30 to 20:15 UTC, and 8 to 12 km altitude; Fig. 5d). This plume is reproduced by both CAMS model configurations, and trajectories indicate central South America as source of these air masses (see Johansson

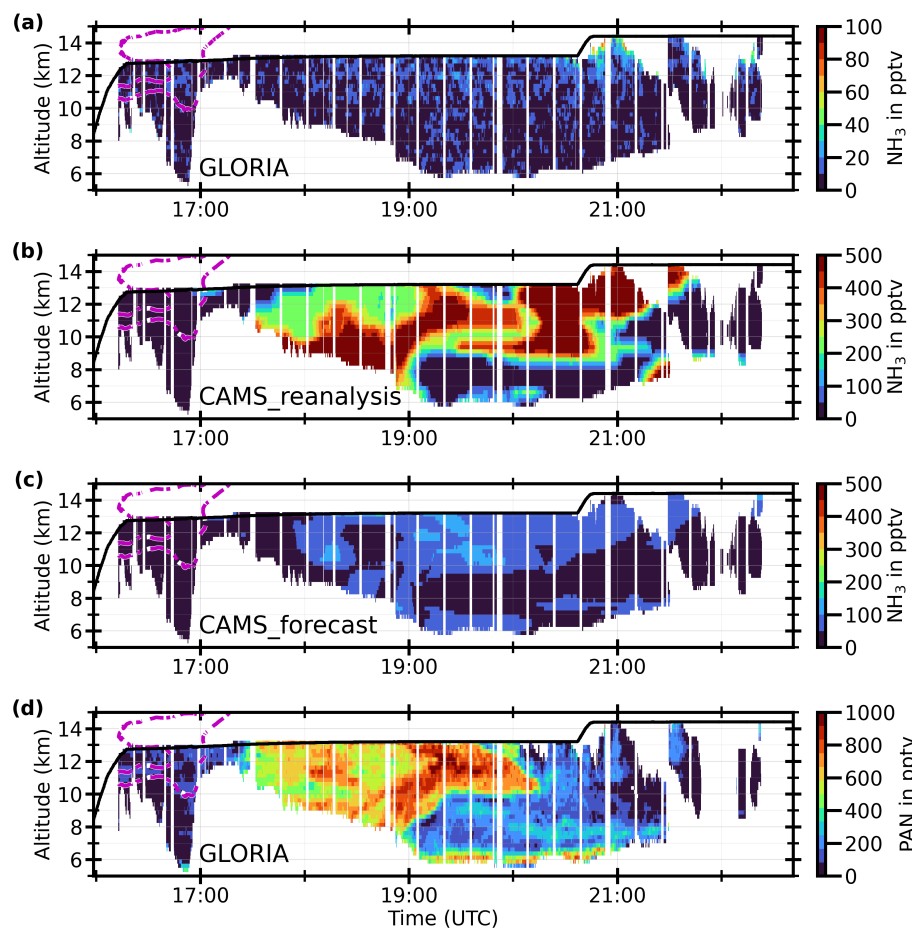

**Figure 5.** Same as Fig. 4, but for flight on 7 October 2019 from Buenos Aires, Argentina, to Sal, Cape Verde.

et al., 2022 and supplementary figures). Further, background PAN VMRs between 19:00 and 21:00 UTC are up to 350 pptv and thus considerably larger than tropospheric background concentrations in the beginning of the flight, which indicates further pollution outside the major PAN plume. $NH_3$ as measured by GLORIA (Fig. 5a), does not show any plume-like structure, but

background VMRs below 30 pptv. Again, close to a possible aerosol or cloud contamination at 21:00 UTC and 13 km altitude, larger $NH_3$ VMRs up to 70 pptv are observed. As mentioned for measurements of SouthTRAC flight on 8 September 2019, this $NH_3$ enhancement might be an artifact due to the challenging conditions for the retrieval in the presence of clouds and aerosols.

In contrast to the observations, the CAMS reanalysis simulates large amounts of $NH_3$ of more than 1500 pptv. Parts of the

200 simulated $NH_3$ plume resemble the large nose-like PAN plume, as measured by GLORIA. Other parts of the plume include the air masses of the elevated background PAN VMRs, and also the region of enhanced measured $NH_3$, but more than an order of magnitude higher than observed. The CAMS forecast does not simulate as high $NH_3$ VMRs as the reanalysis, but

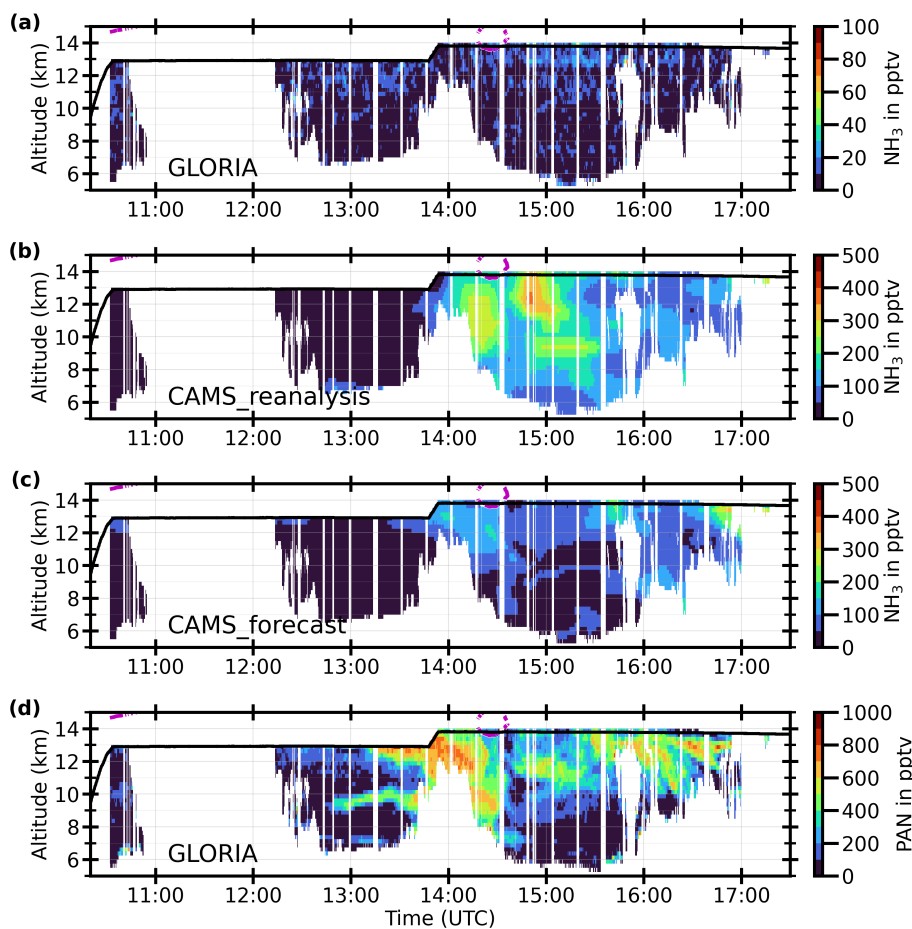

**Figure 6.** Same as Fig. 4, but for flight on 4 November 2019 from Sal, Cape Verde, to Buenos Aires, Argentina.

still maximum VMRs up to 150 pptv, which is considerably more than measured by GLORIA. The structure of maximum simulated $NH_3$ VMRs coincides with measured PAN maxima and thus indicates that the CAMS forecast also simulates large

$NH_3$ VMRs within biomass burning plumes.

   Similar to both presented SouthTRAC flights, the GLORIA observations on 4 November 2019 do not show enhanced $NH_3$ concentrations, despite PAN plumes measured in these air masses (see Fig. 6a,d). Between 14:30 and 16:00 UTC, small enhancements of $NH_3$ are observed at 13 km altitude, just below flight altitude. Otherwise, $NH_3$ VMRs are below 20 pptv. Again, CAMS reanalysis and forecast largely overestimate $NH_3$ for this flight: CAMS reanalysis (Fig. 6b) simulates up to

400 pptv of $NH_3$ at measurement times with enhanced PAN observations. However, simulated $NH_3$ structures are not directly comparable to the observed PAN distributions. Still, simulated PAN patterns, in particular at 15:00 UTC, are similar to this $NH_3$ distribution (see supplementary Fig. S5). This indicates an overall displacement of the simulated plumes compared to the

GLORIA observations. CAMS forecast (Fig. 6c) again shows lower $NH_3$ concentrations compared to the reanalysis, but still values of up to 150 pptv. In this case, simulated $NH_3$ patterns resemble more the structures observed in PAN.

## 3.3 Discussion

The GLORIA $NH_3$ observations show very different situations for the atmosphere above the Asian Monsoon and above the South Atlantic. While GLORIA measured plumes of strongly enhanced $NH_3$ VMRs in the upper troposphere within the Asian summer Monsoon, biomass burning plumes above the South Atlantic do not show detectable enhancements of $NH_3$. Comparisons with both CAMS models for the StratoClim measurements during the Asian monsoon show reasonable agreement
in quantitative terms as well as in the location of the plumes (within the expected performance of the model). In particular the better representation of the major plume during the flight on 31 July 2017 in the CAMS forecast (compared to the CAMS reanalysis) may be due to the higher horizontal resolution and upgraded model properties of the forecast (see Sec. 2.2.2).

For the SouthTRAC flights above the South Atlantic, CAMS reanalysis and forecast both simulate $NH_3$ plumes of several hundred pptv, which are correlated with biomass burning plumes, but which are not observed by GLORIA. The simulated
biomass burning plumes (in particular the PAN plumes) show reasonable agreement with the GLORIA PAN measurements, which excludes a mismatch of the plume locations between model and observation as reason for the disagreement in $NH_3$. In particular, the spatial resolution employed by CAMS has shown to be useful to reproduce the biomass burning plumes that are discussed in this work.

This difference in the ability of CAMS models to reproduce measured $NH_3$ VMRs may have various reasons: Sources for
$NH_3$ may be of different quality for different regions on Earth and also for different emission types. While the enhancements of $NH_3$ measured in the Asian Monsoon on 31 July 2017 likely stem from anthropogenic (agricultural) activities (see Supplementary figures 10-11 of Höpfner et al., 2019), plumes above the South Atlantic, in which CAMS models show enhanced $NH_3$, are likely to come from biomass burning events, as it was shown by (Johansson et al., 2022) by employing backward trajectories and analyses of different pollutants for these air masses. These two types of emissions, anthropogenic and biomass
burning, are prescribed by different emission data sets in CAMS model simulations (see Sec. 2.2), and both data sets may vary in their accuracy and precision regarding $NH_3$.

When comparing the amount of $NH_3$ in the lower atmosphere near the surface (Supplementary figures S6-S7), the CAMS reanalysis dataset shows comparable values in the order of 10 ppbv in the Asian Monsoon over Northern India/Pakistan by end of July 2017 as well as over the burning regions in the Amazonian in October 2019. The concentrations in the Monsoon
region are in broad agreement with the total column amounts of $NH_3$ as observed by IASI (Clarisse et al., 2023). Regarding the biomass-burning cases, the near-surface concentrations of the model appears consistent with observations of $NH_3$ within fire plumes of tens to few hundreds of ppbv (Tomsche et al., 2023). However, IASI total column amounts in the region of the South American plume-origin in autumn 2019 are by a factor of around 5 less abundant compared to the ones inside the monsoon. This would indicate an overestimation of the model with respect to the Amazonian fire sources. In addition, $NH_3$ emissions
of fire plumes are highly variable and strongly depend on factors such as fire size, soil composition, fuel composition, or fire weather (e.g., Tomsche et al., 2023).

Further, atmospheric sinks, such as wet deposition or formation of ammonium ($NH_4^+$)-containing aerosols, may be underestimated by CAMS model configurations within biomass burning plumes above the South Atlantic. In order to evaluate the ability of the model to reproduce the formation of $NH_4^+$ aerosols, we compared the GLORIA AN for the SouthTRAC flights with CAMS forecast ammonium aerosol (Supplementary figures S8-S10). Similar to the $NH_3$ comparisons, AN is not notably enhanced in the GLORIA measurements (beside small enhancements, which correlate with enhanced cloud contamination), but it is considerably enhanced in CAMS forecast. Since the formation of AN is expected to happen in the upper troposphere, the absence of AN in GLORIA data, together with the presence of AN in CAMS forecast suggest that the model already transported too much $NH_3$ into the upper troposphere for the biomass burning plumes. This overestimation of AN is in line with recent comparisons of different CAMS model configurations, where older configurations (such as the 2019 CAMS forecast configuration) tend to overestimate ammonium (Rémy et al., 2024).

In order to find hints of missing wet deposition, GLORIA cloud index is compared to CAMS forecast cloud fraction (Supplementary figures S8-S10). For all three examined SouthTRAC flights it seems like the model underestimates the presence of clouds in the measured air masses. However, this comparison is potentially misleading, since the GLORIA cloud index could potentially be influenced by aerosols, or by clouds, which are along the line of sight, but not necessarily on the tangent point, on which the CAMS forecast cloud fraction was sampled.

In addition, it is still under discussion, how $NH_3$ is efficiently transported into the upper troposphere in the Asian Monsoon without being washed-out by the strong precipitation within thunderstorms. For example, Ge et al. (2018) explain large $NH_3$ VMRs by deep convective vertical transport of $NH_3$ being dissolved in liquid cloud droplets, from which $NH_3$ would be released during the freezing process of the cloud particles. However, this transport mechanism may be imitated by the CAMS models by a lower wash-out, which is helpful for the Asian Monsoon, but not for biomass burning above the South Atlantic.

## 4 Conclusions

In this work, we have tested different CAMS model configurations with respect to their capability to simulate $NH_3$ mixing ratios in the upper troposphere. Knowledge about $NH_3$ in the atmosphere is important due to the influence of $NH_3$ on aerosol formation in an altitude region strongly influencing Earth's radiative budget.

We have selected two general scenarios, where enhanced concentrations of $NH_3$ can be expected to be transported from ground sources to high altitudes: the Asian monsoon, and strong biomass burning in South America and Africa. In both cases, the CAMS model configurations simulated elevated amounts of $NH_3$ in the upper troposphere.

Comparing these model data with observation of $NH_3$ obtained by the GLORIA infrared remote sounding instrument during two aircraft measurement campaigns reveals that the model configurations reproduce broadly well the observations of elevated $NH_3$ concentrations within the Asian summer monsoon. However, they strongly overestimate $NH_3$ within the biomass burning plumes in the mid-upper tropical troposphere, where no clear enhancement is visible in our measurements.

Clarifying the reason for this model overestimation is beyond the scope our analysis. However, we suggest that the emission of $NH_3$ through biomass burning might be overestimated. It is known that $NH_3$ emissions are challenging to simulate due to

280 its large variability, which is among others influenced by fire size, soil and fuel composition, and fire weather. Still there exist large uncertainties about the losses of $NH_3$ through different processes like particle formation and wash-out on its way from its source region to the measurement location in the upper troposphere. By comparing measurements of GLORIA AN with an estimation of CAMS forecast AN, we suggest that conversion of $NH_3$ into AN is not responsible for the overestimation of $NH_3$ by the model.

In terms of measurements, $NH_3$ is strongly undersampled especially at higher altitudes of our atmosphere, which means that there is a need for more vertically resolved observations. Here we have shown, that by using vertically high resolved observation from spectroscopic infrared limb-observations, it is possible to judge on the skills of atmospheric models to describe this trace gas correctly. With hopefully upcoming satellite missions, like ESA's Earth explorer 11 candidate CAIRT (Changing-Atmosphere Infra-Red Tomography Explorer; Sinnhuber et al., 2023), it will be possible to get daily global and altitude-290 resolved vertical distribution of $NH_3$ from the mid-troposphere to the lower stratosphere. In combination with infrared nadir sounders, like IASI, it will be possible to gain much more insight in the processes governing the distribution of $NH_3$ in our atmosphere.

*Data availability.* GLORIA measurements are available in the database HALO-DB (https://halo-db.pa.op.dlr.de/mission/101 and https://halo-db.pa.op.dlr.de/mission/116) and is available on the KITopen repository (https://doi.org/10.35097/btwqkKRszRMeSLTm). The CAMS 295 model data is available from the Copernicus Atmosphere Data Store (https://ads.atmosphere.copernicus.eu). IASI $NH_3$ v4 columns are available online: https://iasi.aeris-data.fr/NH3/

*Author contributions.* SJ initiated the study, performed the analyses, and wrote the manuscript together with MH. GW, MH, JU, AK, NG, PP, and SJ performed the GLORIA data processing. FFV, EK, TG, CP, GM, HN, TN and coworkers operated GLORIA during the StratoClim and SouthTRAC campaigns. MH, MR, JU, BMS, WW and SJ participated in scientific flight planning during the campaigns. VH provided 300 insights into the CAMS simulations. All authors commented on and improved the manuscript.

*Competing interests.* The authors declare that they have no conflict of interest.

*Acknowledgements.* We gratefully thank the SouthTRAC coordination team, in particular DLR-FX for successfully conducting the field campaign, and local support from Argentina and Chile. For the successful campaign from Kathmandu, we thank the StratoClim coordination team as well as the Geophysica team, and local support from Nepal. The results are based on the efforts of all members of the GLORIA 305 team, including the technology institutes ZEA-1 and ZEA-2 at Forschungszentrum Jülich and the Institute for Data Processing and Electronics at the Karlsruhe Institute of Technology. The StratoClim campaign was supported by the European Community's Seventh Framework Programme (FP7/2007–2013) under grant no. 603557. The SouthTRAC measurement campaign was supported by the German Science

Foundation (Deutsche Forschungsgemeinschaft, DFG) under the Priority Program HALO SPP 1294. Within the HALO SPP this work was supported by project grant HO 4120/4-1. In addition the participating Helmholtz institutes contributed to the HALO operation costs based on their programme-oriented funding We thank ECMWF for providing CAMS data. Contains Copernicus Atmosphere Monitoring Service Information 2023. Neither the European Commission nor ECMWF is responsible for any use that may be made of the information it contains. We thank AERIS for providing IASI $NH_3$ data, and we particularly appreciate helpful comments from Lieven Clarisse. We acknowledge support by the KIT-Publication Fund of the Karlsruhe Institute of Technology.

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
