# Peer review of "Ammonia in the UTLS: GLORIA airborne measurements for CAMS model evaluation in the Asian Monsoon and in biomass burning plumes above the South Atlantic"

_EGUsphere, 2024_

## Author Comment (AC1)

**Response to referee 1:**

Comments of the referee are repeated in **bold**, changes in the manuscript are highlighted in *italics*.

**General comments**

**The preprint from Johansson et al. presents NH3 measurement from the GLORIA instrument abord HALO and compares the data to CAMS model simulations (reanalysis and forecast) for two distinct situations in the upper troposphere. First, NH3 in the Asian monsoon and second biomass burning event from South America over the Southern Atlantic. While the model represent well NH3 in the Asian monsoon, it overestimates NH3 in biomass burning events into the upper troposphere over the Southern Atlantic. In contrast to NH3, PAN is simulated well with the model. Johansson et al. suggest some hints and reasons, why the model has difficulties in NH3 representation for some NH3 sources, but a deeper analysis on the reasons is out of the scope of the paper.**

**The subject of NH3 in the upper troposphere, especially with its potential indirect contribution to the Earth's radiation budget and the difficulties in the representation in models fits well in the scope of ACP.**

**Overall the paper is easy to read and the results are presented in a logically structured manner. It has only minor issues (presented below) and I recommend publishing the paper.**

We thank the referee for these positive and encouraging words and for improving our manuscript with specific and thoughtful comments.

**Specific comments**

**Title: I suggest to either write "ammonia" or "NH3" in the title, but not both.**

Agreed and changed to "ammonia".

**Figure 4-6: I suggest to add the information of the take off and landing base for each flight in the Figure description. That helps to better understand the flight direction and thus the latitudinal location as it is hard to differentiate the three flight in Figure 1b.**

We thank the referee for this suggestion. We added origin and destination airports in the figure descriptions. For completeness, we also added this information for Figure 2.

**Figure 4: I can only identify one PV line. Is it the 2 PVU or 4 PVU line? The same is valid for Fig 5 and Fig 6. I suggest either to add a label in the plot or clarify in the caption, which line is visible.**

We agree that the -4 PVU line is only visible during a very small section of the flight. We added to the caption: *Due to tropopause heights, which are partially located above flight altitude, these magenta lines are not always present, and sometimes only the ±2 PVU line is present.*

**L145-147: How is the horizonal distribution of NH3 over the Indian subcontinent? Could different sources on the ground with different NH3 emissions have lead to two distinct air masses characteristics east and west of the flight tracks?**

Our GLORIA measurements suggest that the horizontal distribution of NH3 in the UTLS is very variable. We add to the manuscript: *Hence, our GLORIA measurements indicate strong horizontal variability of NH$_3$ in the Asian Monsoon upper troposphere.*

**L160-162: The presents of clouds explains the reduced quality of the NH3 retrieval, but could they also be a hint of NH3 loss and transformation processes (e.g. particle formation), leading to reduced NH3 mixing ratios? How are clouds represented in CAMS and could it be a hint of missing scavenging effects for NH3??? Are there any information on the ammonium (NH4+), thus the particle phase available, either for Gloria or model data, which could give further ideas on the deviation between model and measurements.**

We added supplementary figures S8-S10, comparing GLORIA and CAMS ammonium nitrate, together with GLORIA cloud information (cloud index) with CAMS forecast cloud fraction. We added a paragraph to the "discussion" section, to include these new supplementary figures in the manuscript.

**L214: You state that NH3 over the South Atlantic is likely from biomass burning. Could there also be a contribution from livestock and fertilizers?**

Johansson et al. (2022) showed that the air masses, in which CAMS shows enhanced NH3, are connected to biomass burning events. This is also reflected in the enhancements of PAN (lowermost panel in figures 4-6), which is connected to pollution (i.e. biomass burning), but not with livestock and fertilizers. We rephrased the paragraph to emphasize the previous studies of Johansson et al. (2022).

**L224: NH3 is very variable in biomass burning, depending on a lot of factors, e.g. fire size, soil composition, fuel composition, fire weather etc. I suggest to mention some of the reasons, why NH3 is hard to simulate, especially from BB sources, at least at some point, see also L243.**

We thank the referee for this comment. We added to the manuscript:
*In addition, NH$_3$ emissions of fire plumes are highly variable and strongly depend on factors such as fire size, soil composition, fuel composition, or fire weather (e.g., Tomsche et al., 2023).*

**L225-231: As CAMS seems to reproduce the large scale Asian Monsoon transport of NH3 well in comparison to biomass burning. Could that be a size problem of CAMS, as the resolution for small scale phenomena is not sufficient enough?**

Since CAMS seems to succeed to reproduce the PAN within biomass burning plumes, Johansson et al. (2022) concluded that dynamics are well captured by the model. We mentioned this in the second paragraph of the discussion, but we tried to further emphasize this in the revised manuscript.

**L237: You state strong biomass burning in South America and Africa. I could not find any other statement concerning Africa as a source region in the draft for the flight during SouthTrac. Is Africa a potential source region, which need to be taken into account?**

We agree that Africa as potential source region (as indicated by Johansson et al. (2022) and their trajectory analyses) should be mentioned earlier than in the conclusions section. We added Africa as source region of the biomass burning plumes in the section describing SouthTRAC flight on 8 September 2019. However, CAMS only showed enhancements of NH3 within the South American plumes.

**L243: I agree that clarifying the reasons for NH3 overestimation is beyond the scope of the presented analysis. Nevertheless, I suggest to mention some more difficulties of NH3 in simulations. Some suggestions are: NH3 is very variable in biomass burning, depending on a lot of factors, e.g. fire size, soil composition, fuel composition, fire weather etc. The chemical aging of NH3 in BB plumes can be fast. BB emit moisture and particles, thus accelerating transformation and scavenging processes, leading to NH3 loss. Consequently, it is not further transported in comparison to PAN from South America to the Southern Atlantic upper troposphere.**

We thank the referee for these further suggestions, which we now included in the discussion and conclusions section.

**What about NH4 in model and Gloria data (see also L160-162)**

We now added ammonium nitrate comparison figures to the supplement and briefly discuss these comparisons (see also response to comment on L160-162). Further, we mention these findings now in the conclusions.

**L247: I strongly agree that NH3 is undersampled.**

We thank the referee for supporting our statement.

**Technical corrections**

**L12: "pollution gases, such as peroxyacetyl nitrate (PAN) show the ability" -> "pollution gases, such as peroxyacetyl nitrate (PAN), show the ability"**

Done.

**L13: Suggestion "CAMS models to reproduce the biomass burning plumes above the South Atlantic in general." -> ""CAMS models to generally reproduce the biomass burning plumes above the South Atlantic."**

Agreed.

**L15: Suggestion "… different accuracy for biomass burning and agricultural sources in the Asian Monsoon" -> "… different accuracy for biomass burning in comparison to agricultural sources in the Asian Monsoon"**

Agreed.

**L28: ".. local air quality, and for climate through …" -> ".. local air quality and for climate through …" also L42, L54**

We have no native speakers in our team of authors, but to our understanding the oxford comma is optional. However, language copy editing will be applied to the manuscript before publication in ACP, where such issues are typically handled. For that reason, we did not remove the comma as it was suggested here.

**L38: "or AIRS (Atmospheric Infrared Sounder) global…" -> "or AIRS (Atmospheric Infrared Sounder), global…"**

Done.

**L47: "In the present work we utilize…" -> "In the present work, we utilize…"**

Agreed.

**L64: "… sampling up to 0.0625 cm−1, and compensation of aircraft movements and active targeted…" -> "… sampling up to 0.0625 cm−1, compensation of aircraft movements, and active targeted…"**

Since the compensation of aircraft movements and the active selection of target air masses is considered as one feature of GLORIA, we would like to keep the original formulation, and wait for suggestions from language copy editing.

**L79, L81: "… 1.3 km, and estimated…" -> "… 1.3 km and estimated…" similar L92, L94, L100, L106, L110 (2x), L174, L184**

**L85: "…, and second, the CAMS .." -> "… and second, the CAMS .." similar L86**

Again, we would like to keep the oxford comma, unless language copy editing suggests otherwise.

**L90/91: "2003 and 2021, and uses 60 vertical levels between 0.1 and 1000 hPa, and a horizontal…" -> "2003 and 2021, uses 60 vertical levels between 0.1 and 1000 hPa and a horizontal…"**

Agreed.

**L149: Suggestion "and supplementary figures 1-2)." -> "and supplementary Fig. S1 and Fig. S2)."**

Agreed.

**L177: Suggestion "NH3 as measured by GLORIA (Fig. 5a) instead, does not…" -> "NH3 as measured by GLORIA (Fig. 5a), does not…"**

Agreed.

**L192: "(Figs6b)" -> "(Fig. 6b)"**

Done.

**L195: "… simulated plumes, compared to…" -> "… simulated plumes compared to…"**

Agreed.

**L200: "enhancements of ammonia." -> "enhancements of NH3." Also L234, L253**

Done.

We thank the referee for the detailed comments and for improving our manuscript.

**Response to referee 2:**

Comments of the referee are repeated in **bold**, changes in the manuscript are highlighted in *italics*.

**This study presents valuable model-measurements comparisons of NH3 in the upper troposphere and lower stratosphere, focused on the biomass burning event and the Asian summer monsoon over two specific locations. Large enhancement of NH3 was observed in the upper troposphere near the Asian monsoon region but not over the biomass burning plumes. The CAMS model simulation shows enhancement of NH3 in both the regions. Overall, the results are well organized and presented with sufficient scientific evaluation. However, there are some weaknesses that could be improved to make this work more impactful.**

We thank the referee for this statement and for improving our manuscript with comments.

**General Comments:**

**1. The goal of this study should be mentioned clearly. Is it to evaluate the CAMS model? Or to understand the differences in CAMS reanalysis and real time products? Is the end goal to improve the model performances?**

We now mention more explicitly in the end of the introduction that we aim to evaluate upper tropospheric NH$_3$ concentrations in two configurations of the CAMS model (see also response to specific comments).

**2. Both NH3 and PAN are both emitted from biomass burning and discussed in the manuscript together. I am not sure how to they are related to each other. Can we expect the same enhancement in them due to biomass burning? What are their lifetimes?**

We added a paragraph to the introduction to motivate the usage of PAN as an established tracer for biomass burning.

**In all the figures, GLORIA measurements of NH3 and PAN are compared with the CAMS NH3. What does PAN simulated from CAMS look like for these cases?**

Supplementary figures S1-S5 show comparisons between GLORIA and CAMS for PAN for all discussed flights. Further there are references in the manuscript to previous work, which extensively discussed PAN evaluation of CAMS reanalysis based on these GLORIA measurements (Johansson et al., 2020, 2022). We think that this point is already sufficiently discussed in the manuscript.

**3. The CAMS model performance is evaluated for the two specific events, Asian monsoon and biomass burning. It would be valuable to include some information about how CAMS performs on climatological perspectives. Can satellite measurements of NH3 be compared with the CAMS model outputs?**

Unfortunately, there are no satellite measurements of NH$_3$ in the upper troposphere available for the time range of this work. As discussed in the manuscript, there are total column measurements available from nadir infrared sounders, which, however, collect most of their signal from surface levels. Between 2002 and 2012, MIPAS on the Envisat satellite measured 3-monthly and 10° x 10° (latitude x longitude) mean NH$_3$ in the upper troposphere, but again only above the Asian Monsoon region (Höpfner et al. 2016).
We agree that it would be very helpful to have satellite measurements of NH$_3$ in the upper troposphere and we are thus looking forward to the proposed CAIRT mission, which would enable regular measurements.

**4. Both the CAMS reanalysis and forecast products were compared in this study. Is either one expected to be better than the other?**

We expanded the model description part of the manuscript in order to discuss the advantages and disadvantages of the CAMS forecast and CAMS reanalysis configurations:
*For this work, we compare GLORIA data to both CAMS configurations to achieve two goals: First, we want to compare GLORIA $NH_3$ obtained during two different campaigns 2017 and 2019 with a consistent data set that did not change for the time ranges of both campaigns. For this purpose, the reanalysis configuration is suited best. Second, we want to compare GLORIA $NH_3$ to the most recent CAMS configuration that is publicly available, which is the CAMS forecast configuration, in order to check for improvements for more recent model versions.*

**5. For all the figures, the timing of enhancement in the measurements and the model seems to be different. Therefore, it is hard to draw conclusions based on these figures. Do we expect the timing to be closer?**

For all comparisons, we used the best temporal resolution available (3 h). However, we do not think that the temporal resolution is causing the differences in $NH_3$, since the biomass burning plume structures of PAN were nicely reproduced by both CAMS model configurations in this work (see also supplementary Figs. S1-S5).

**6. For the model-measurements differences, only some speculations were made without any evidence. Are there any supporting material to show, for example, that the loss process of HN3 is underestimated?**

We added as further supporting material for the biomass burning flights GLORIA retrievals of solid ammonium nitrate, and compared these distributions to CAMS forecast ammonium aerosol. Since the solid ammonium nitrate was not measured by GLORIA, but simulated by CAMS forecast, we now suggest that the underestimated loss process is not the conversion to solid ammonium nitrate in the upper troposphere.
For other loss processes (and also the sources), we do not have more evidence than we already have shown. As already mentioned in the manuscript, further model investigations to fully explain the discrepancies is out of scope for this work.

**Specific Comments:**

**L5 (Abstract) – within the Asian monsoon -> either Asian monsoon anticyclone or Asian monsoon region?**

The StratoClim campaign was conducted in the Asian Monsoon region and performed flights within the Asian Monsoon Anticyclone. In order to emphasize the uniqueness of these StratoClim flights, we decided to mention the flights within the Asian Monsoon Anticyclone.

**L7 (Abstract) – We compare… model. -> We compare…model to evaluate its performances.**

Agreed.

**L8 (Abstract) – both reanalysis and forecast model were used here. Why?**

We think it would be too much to explain the choice of model configurations in detail in the abstract, but as mentioned for the general point 4, we decided to expand the model description to answer this question.

**L15 (Abstract) – What does 'different accuracy' mean? Is it higher or lower? Or better?**

Thanks for pointing this out. We specified that we suggest that the biomass burning emissions are of lower accuracy.

**L24 (page 2) – Is this due to warming climate?**

We specified in the text that increased emissions of $NH_3$ are expected *due to increased livestock, usage of fertilizers, and combustion.*

**L33 (page 2) – Here 'quasi non' might mean 'close to zero'. I would like to know if it exists or not.**

We changed this formulation to: *very sparse measurements with high uncertainties*

**L47 (page 2) – This sentence is important in stating the goal of this work. Is it to analyze the model performance or describe the distribution? Or it could be to evaluate the model under two different conditions?**

We clarified that the goal of the article is to evaluate CAMS in two different atmospheric conditions.

**L66 (page 3) – What does 'mostly interferograms…used' mean?**

Sorry for the confusion: We removed the word "mostly", to make clear that interferograms with full MOPD are used, unless they are not available.

**L79 (page 4) – '…adjusted slightly' can be explained explicitly.**

We added the explicit micro windows for both MOPDs in the new supplementary table S1 and refer to this table in the main text.

**Section 2.2.1 (Page 4) – Here, the CAMS performance is stated for PAN. Is there a direct relationship between PAN and NH3 other than both are biomass burning tracers? Are there any studies focused on NH3?**

The performance of CAMS for PAN in biomass burning plumes is important to know, since it shows that the dynamics are simulated well within CAMS. The relationship between PAN and NH3 is, indeed, that these both can be emitted during biomass burning processes.
Further, we added to the manuscript: *To our knowledge, there are no studies available evaluating $NH_3$ for CAMS reanalysis. However, since $NH_3$ and PAN both can be emitted from biomass burning events, it is important to know that CAMS is able to reproduce biomass burning PAN plumes.*

**Figure 2 (page 6) – The color bar is saturated at 1000 pptv. However, the values in Fig. 2a & 2b could be much higher than 1000 pptv.**

We now mention the maximum VMRs in the figure captions. We decided to keep the color bar range saturated at 1000 pptv, in order to not lose focus on structures with lower VMRs.

**L145 (page 7) – 'This is remarkable…changed'. -> I am not sure what this sentence means.**

We rephrased this sentence in the revised manuscript.

**L162 (page 9) – Were there cloud or aerosols? Fig. 4a (also 5a) looks likes very noisy to me.**

The mentioned line in the original manuscript was discussing filtered data, which is thus not shown in the figures. However, as suggested by referee 1, we added plots showing cloud information (supplementary figures S8 – S10).

**L243 (page 12) – 'There are..hints…' -> Can this be shown or explained further?**

As mentioned in the manuscript, it is out of the scope of this work to perform sensitivity studies, which could show if the accuracy of the emissions is causing the mismatch between observation and model. However, we rephrased this sentence to make clear that the emissions are suggested as error source by our work.

We thank the referee for the comments and for improving our manuscript.